# OpenReview forum: "Impact of LLM Alignment on Impression Formation in Social Interactions"
_colmweb.org/COLM/2025/Conference — COLM 2025_

### Official Review · Reviewer_tUeP · 2025-04-20

**Rating:** 6
**Confidence:** 2
**Ethics Flag:** 1

**Summary:**

The paper uses Affect Control Theory (ACT) to assess gender impressions in LLMs, revealing that models primarily rely on actor identity and exhibit limited sensitivity to behavior and recipient context compared to ACT's predictions.

**Reasons To Accept:**

1. The article introduces the empirical method ACT as a theoretical framework for LLM social impression for the first time, providing a new perspective for understanding biases in LLMs.

2. It conducts thorough experiments, verifying the relationships among multiple variables through detailed statistical analysis.

**Reasons To Reject:**

This paper focuses primarily on quantifying the LLM impression by ACT, but some of its conclusions lack sufficient evidence to be fully persuasive.

1. The experimental results show that even after post-training, LLMs still exhibit gender biases, presenting an interesting contrast to the expectation that alignment reduces bias. I think the reason is not that all the post-training is for debiasing. So if you stick to your conclusion, which you make in lines 369-370, you need to strengthen the persuasiveness by comparing the results from ACT with post-training studies specifically designed for debiasing. For example: MBIAS: Mitigating Bias in Large Language Models While Retaining Context (Raza et al., WASSA 2024)

2. The ACT corpus used for the experiment is sourced only from the U.S., so I have a concern about whether, for an empirical method, corpora in different languages would lead to significant differences. As you discussed in line 364, more experiments on DeepSeek (or other Chinese models) and Chinese corpus are strongly recommended.

3. In Table 1, there are many values that are insignificant, which makes me feel a little bit unfair in the prompt. The ACT method is designed right for the 'actor', 'behavior', and 'recipient'; however, in the prompt, you didn't tell the LLMs to consider it in this way. Intuitively, this may lead to different results. How do you balance this 'fairness'?

4. You conclude that 'LLM impressions differ substantially from ACT, which seems too absolute. I suggest specifying that these are gender impressions. As you mentioned in the future work section, there are more aspects to be explored through ACT.

5. You propose a new metric to evaluate the bias, which I think is necessary to use a subsection 'metrics' to clearly introduce this part. I suggest a brief introduction to each of your abbreviations in table 1, for me, I don't get 'pr' at first. I suggest showing part of your dataset in the appendix to help readers understand ACT and VAD, including giving out an example of how to understand the values and their meanings.

---

> ### Author Response · Authors · 2025-06-01
> **Response to Reviewer tUeP’s Comments**
>
> Thank you for your constructive feedback.
>
> **Regarding post-training debiasing:** This is an excellent suggestion. Following your feedback, we conducted an initial analysis comparing **Tulu-3-8B-SFT** (trained with safety/bias data) and **Tulu-3-8B-SFT-no-safety-data**. We measured gender impression across all scenarios. As shown below, the male-leaning bias persists across both models, suggesting that safety-aligned data did not substantially shift gender impression:
>
> | Model                  | Female Probability M (SD) | Male Probability M (SD) |
> |------------------------|---------------------------|--------------------------|
> | Tulu-3-8B-SFT          | 0.280 (0.216)             | 0.720 (0.216)            |
> | Tulu-3-8B-SFT-no-safety| 0.278 (0.206)             | 0.722 (0.206)            |
>
> Although post-training is not always explicitly designed for debiasing, as detailed in Appendix A, all models we evaluate have undergone safety and bias alignment procedures. For instance:
> - LLaMA evaluates safety across toxicity, truthfulness, and bias [1].
> - Tulu includes training data focused on social stereotypes and discrimination [2].
> - DeepSeek R1 incorporates an RL stage to improve helpfulness and reduce harmful or biased outputs [3].
>
> We will highlight this as a promising direction for future work: using our ACT-based framework to evaluate the impact of targeted debiasing strategies.
>
> **Regarding cultural scope:**  You raised an excellent point about culture, which we also emphasize in Section 5. Our initial focus on the English-language ACT dictionary stems from the fact that LLMs are primarily trained on English data (e.g., >90% for LLaMA [4]) and tend to align more closely with U.S. cultural norms. That said, ACT provides dictionaries from many cultures, including Chinese [5], offering a strong foundation for exploring cross-cultural variation.
>
> Prior ACT research suggests that collectivist cultures, such as China, may place greater weight on behavior in impression formation [6]. However, we find no evidence for this in our preliminary analysis of the DeepSeek model, which exhibits similar behavior compared to other models. Recent work further supports this pattern, showing that both China-origin and U.S.-origin models tend to align more closely with U.S. cultural norms than with Chinese ones [7].
>
> Following your feedback, we are actively analyzing gender impression predictions using the Chinese ACT dictionary. If preliminary results are available in time, we will include them in the rebuttal; otherwise, they will be included in the final version.
>
> **Regarding prompt fairness:**  We used multiple prompt phrasings to improve generalizability (detailed in Appendix B). Agreement across prompt variations is reported in Appendix E, showing medium to high agreement. Our prompts were designed to mirror sentence formats used in human ACT surveys to ensure alignment with prior impression-formation studies.
> In response to your comment, we tested a more explicit prompt labeling ACT roles:
>
> > "A champion insults servant. Considering this ACTOR-BEHAVIOR-RECIPIENT event, what is the gender of the champion (ACTOR)? The gender of the champion is more likely to be..."
>
> Repeating Table 1 with this phrasing on LLaMA-3-Instruct yielded nearly identical results:
>
> | Role    | Effect Size |
> |-------------|-------------|
> | Actor       | 0.17        |
> | Behavior    | 0.00        |
> | Recipient   | 0.00        |
>
> Thus, the model's gender inference still depends almost entirely on the actor's identity.
>
> **Regarding generalization of conclusions:**  We will revise the title to reflect the gender-specific scope—**“Impact of LLM …: A Case Study of Gendered Impressions”**—and update the manuscript to avoid overgeneralizations.
>
> **Clarifying metrics and abbreviations:**  In the final version, we will clearly introduce all abbreviations and metrics (e.g., “pr”), along with sample ACT dataset entries and VAD values.
>
>
> We believe these revisions directly address your concerns. Based on your and other reviewers' feedback, we have added new evaluations, including internal mechanism analysis, prompt variation, and safety-trained model comparisons, to strengthen the paper. We welcome any further comments during the discussion period and are happy to provide additional clarifications if needed.
>
> **References:**
>
> [1] Touvron et al, *LLaMA 2: Open Foundation and Fine-Tuned Chat Models*, 2023.
> [2] Lambert et al, *Tulu 3: Pushing Frontiers in Open Language Model Post-Training*, 2024.
> [3] Guo et al, *DeepSeek-R1: Incentivizing Reasoning via Reinforcement Learning*, 2025.
> [4] Kew et al, *Turning English-centric LLMs Into Polyglots: How Much Multilinguality Is Needed?*, 2024.
> [5] Smith & Cai, *Mean Affective Ratings of 1146 Concepts by Shanghai Undergraduates*, 1991.
> [6] Zhao, Y., *Modeling Impression Formation Processes Among Chinese and Americans*, 2023.
> [7] Sukiennik et al, *An Evaluation of Cultural Value Alignment in LLMs*, 2025.

---

> > ### Comment · Reviewer_tUeP · 2025-06-11
> >
> > Thank you for the rebuttal. I increased my score.

---

### Official Review · Reviewer_SonJ · 2025-05-06

**Rating:** 7
**Confidence:** 4
**Ethics Flag:** 1

**Summary:**

This paper investigates whether Large Language Models (LLMs) exhibit patterns of impression formation that align with the Affect Control Theory (ACT).
The author(s) introduce the thoretical framework, which posits that impressions are largely determined by the words people use to label situations.
In practice, using different words to call a person, such as “teacher” or “friend”, determines expectations about their character and the actions they are likely to perform.
The paper presents a case study where the author(s) systematically compare LLM output to ACT predictions to evaluate gender impressions and understand whether LLMs approximate human impression formation according to ACT.
Research questions are clearly stated.
The benchmark is a synthetic dataset but its evaluation is unclear.

The author(s) state that
"We utilize cultural meanings from the USA Combined Surveyor Dictionary (Smith-Lovin et al., 2016) to evaluate the cultural sentiments in each generated scenario, aligning with Western social impressions that LLMs are also largely trained on (Cao et al., 2023). This dictionary is created using ratings from participants on 2,402 social concepts"
are there evaluation metrics that can be reported from Cao et al 2023? what are the results of the evaluation of the cultural sentiments in  generated scenarios?

Prompts and examples are in the appendix but to improve clarity should be included in the data section (3.1).
The author(s) report that LLMs demonstrate insensitivity to situational context: in other words, the impression of an interaction is primarily determined by the actor's identity, irrespective of their actions or the recipient of those actions.

Line 91: valence, arousal, dominance should be uppercased.

I think that the author(s) should improve section 5 with a discussion about the potential benefit of their data and findings for bias detection tasks, citing papers like the following:

@inproceedings{dusi2024supervised,
  title={Supervised Bias Detection in Transformers-based Language Models},
  author={Dusi, Michele and Gerevini, Alfonso Emilio and Putelli, Luca and Serina, Ivan and others},
  booktitle={Proceedings of the CEUR Workshop Proceedings, Vienna, Austria},
  volume={3670},
  year={2024}
}


@inproceedings{morales2023automating,
  title={Automating bias testing of llms},
  author={Morales, Sergio and Claris{\'o}, Robert and Cabot, Jordi},
  booktitle={2023 38th IEEE/ACM International Conference on Automated Software Engineering (ASE)},
  pages={1705--1707},
  year={2023},
  organization={IEEE}
}


Moreover, research in affective computing with role play propmting recently revealed that LLMs can replicate human personality in a credible manner, but they do not have a personality themselves.
Including references to that topic in the discussion would greatly improve the appeal of the paper towards a broad research community. Authors should include and cite papers like the following:


@article{celli2025twenty,
  title={Twenty Years of Personality Computing: Threats, Challenges and Future Directions},
  author={Celli, Fabio and Kartelj, Aleksandar and {\DJ}or{\dj}evi{\'c}, Miljan and Suhartono, Derwin and Filipovi{\'c}, Vladimir and Milutinovi{\'c}, Veljko and Spathoulas, Georgios and Vinciarelli, Alessandro and Kosinski, Michal and Lepri, Bruno},
  journal={arXiv preprint arXiv:2503.02082},
  year={2025}
}

@article{matz2024,
  title={The potential of generative AI for personalized persuasion at scale},
  author={SC Matz, JD Teeny, SS Vaid, H Peters, GM Harari, M Cerf},
  journal={Scientific Reports},
  volume={14},
  issue={1},
  pages={4692},
  year={2024}
}

**Reasons To Accept:**

research grounded in theory
clear experimental design
potentially interesting results for a wide audience

**Reasons To Reject:**

unclear evaluation of the dataset

---

> ### Author Response · Authors · 2025-06-01
> **Response to Reviewer SonJ's Comments**
>
> Thank you for your constructive feedback.
>
>
> **Regarding dataset evaluation clarity:**
> We use the USA Combined Surveyor Dictionary (Smith-Lovin et al., 2016), which provides ratings for social concepts—including identities and behaviors—across Valence, Arousal, and Dominance (VAD) dimensions. From this dataset, we extract pre-event values (i.e., fundamental sentiments) and apply empirically grounded regression equations—developed from large-scale ACT surveys—to compute post-event, contextualized impressions, as described in Section 3.1.
>
>
> For example, a scenario like *“a nurse hits patient”* produces a negatively valenced impression of the nurse, derived through ACT-based regressions trained on human data. In our evaluation, we use these ACT-modeled impressions as a proxy for ground-truth human gender impressions.
>
> We acknowledge the reviewer’s excellent observation that ACT-based post-event estimates are themselves modeled approximations. This raises an important future direction: directly comparing LLM outputs with newly collected human impression data to validate or challenge both ACT and LLM predictions. While such empirical validation is beyond the scope of this study, we will incorporate this discussion into the revised manuscript to clarify how our evaluation is grounded and where its limitations lie.
>
>
> **Prompt and example relocation:**
> Representative prompts and generation examples will be moved from the appendix into Section 3.1 for immediate reference.
>
>
> **Discussion enhancements and new references:**
> We will revise Section 5 to link our benchmark to supervised bias-detection tasks, citing Dusi et al. (2024) and Morales et al. (2023), and relate our findings to recent personality-simulation research (Celli et al., 2025; Matz et al., 2024).
>
>
> We look forward to any additional thoughts or discussion you may have during the discussion period.

---

> > ### Comment · Reviewer_SonJ · 2025-06-06
> >
> > Dear authors, thanks for considering the suggestions. In my opinion the paper will be stronger and more readable. I confirm my previous rating.

---

### Official Review · Reviewer_FtGm · 2025-05-12

**Rating:** 6
**Confidence:** 3
**Ethics Flag:** 1

**Summary:**

This paper makes a contribution by applying Affect Control Theory (ACT) to analyze and benchmark LLM behavior in social interaction contexts, especially on dynamic gendered social interactions. The paper is well-written and easy to follow, with a clear narrative structure and well-motivated research questions, and exposes how alignment techniques can amplify or distort biases, informing future fine-tuning protocols and fairness auditing.

**Reasons To Accept:**

* This paper offers a unique investigation into how preference tuning and alignment influence contextual impression formation, providing a fresh perspective in the alignment research area.

* The paper is well-written, with clear exposition and thorough analysis.

**Reasons To Reject:**

* The paper lacks complete explanations of key symbols and notations, which may hinder understanding for readers from the NLP community. For example, it is unclear what "HLH" in line 176 refers to, or what the symbols $p$ and $\eta$ represent in Table 1.

* The experimental setup does not appear to offer significant methodological innovations. Aside from presenting a fresh perspective, the work does not show substantial differences from prior studies such as [1]:

[1] Kotek, Hadas, Rikker Dockum, and David Sun. "Gender bias and stereotypes in large language models." Proceedings of the ACM Collective Intelligence Conference, 2023.

---

> ### Author Response · Authors · 2025-06-01
> **Response to Reviewer FtGm's Comments**
>
> Thank you for your constructive feedback.
>
> **Regarding missing symbol explanations:**
> We have ensured that Table 1 and the rest of the manuscript include clear legends for all symbols—e.g., *p* (p-value), *η* (effect size)—and that all acronyms such as “HLH” are defined upon first mention to improve clarity for NLP audiences unfamiliar with ACT terminology.
>
> **Regarding methodological novelty:**
> Our goal was to introduce insights from decades of research in psychology and social psychology to the NLP community. While gender bias has been widely studied, our work uses it solely as a case study to demonstrate a broader methodological point: behavioral context might well be ignored in LLM evaluations.
>
> - Unlike prior work such as Kotek et al. (2023), which primarily examines subject-recipient bias via coreference patterns, our analysis focuses on Actor–Behavior–Receptor (ABR) interactions. For instance, we ask: *What happens when a nurse engages in a negative act?* Our results show that LLMs still attribute dominant impressions based on identity, regardless of behavior.
>
> - This interaction-focused framing—operationalized through systematic Valence-Arousal-Dominance (VAD)-based categorization—enables us to capture nuances missing in earlier studies, such as the differing impressions formed when a high-status identity (e.g., a doctor) engages in harmful behavior versus when a low-status identity (e.g., a prisoner) performs a prosocial act.
>
> - We further compare impression patterns across alignment stages (post-training, SFT, RLHF), which, to our knowledge, is the first time this has been done across a range of both open- and closed-source models. This allowed us to rigorously test claims that newer LLMs display "surprising" gender associations (e.g., traditionally male-typed roles becoming female-typed) [1–3]. We show that such reversals are largely confined to GPT, while the open-source models we tested continue to reflect male-dominant patterns.
>
> - We also introduce the log probability of next-token generation as a proxy for gender impression and social deflection, enabling us to capture subtle preference shifts in gender impression formation that are not detected by multi-choice or classification-based metrics.
>
> - Lastly, we believe a core contribution of this work is laying the foundation for future research—demonstrating how ACT-inspired benchmarks can uncover not just static biases but the interactional logic (or lack thereof) within LLMs.
>
> We will incorporate these clarifications into the revised discussion to better position our work within the broader landscape of gender bias research.
>
> In light of your feedback, we will clarify our notation and methodological positioning. Based on feedback from other reviewers, we have also added new evaluations, including analysis of internal mechanisms, prompt variation, and safety-trained model comparison. We look forward to any additional thoughts or discussion you may have during the discussion period.
>
> **References:**
> [1] Takemoto. *The moral machine experiment on large language models.* *Royal Society Open Science*, 2024.
> [2] Fulgu & Capraro. *Surprising gender biases in GPT.* *Computers in Human Behavior Reports*, 2024.
> [3] Spillner. *Unexpected gender stereotypes in AI-generated stories: Hairdressers are female, but so are doctors.*, 2024.

---

### Official Review · Reviewer_7PD1 · 2025-05-15

**Rating:** 6
**Confidence:** 3
**Ethics Flag:** 1

**Summary:**

This paper compares Affect Control Theory (ACT) and large language models (LLMs) in the context of impression formation and social interaction, with a particular focus on gender-related impressions. The key finding is that, unlike ACT, LLMs tend to determine the impression of interactions primarily based on the Actor's identity, rather than context or interactional dynamics. The evaluation relies on ABR (Actor–Behavior–Receptor) profiles, and a benchmark corpus was constructed to support this analysis.

**Questions To Authors:**

The finding that impression formation in LLMs is primarily determined by the Actor, with limited influence from Behavior and Receptor, is both novel and important. I encourage the authors to expand their discussion to explore possible connections between this finding and LLMs' internal mechanisms or architecture. Such a discussion could provide valuable insights into why these patterns arise and how they might differ from human impression formation processes modeled by ACT.

**Reasons To Accept:**

The finding that impression formation patterns in LLMs are not highly context-sensitive but are heavily influenced by the actor identity is particularly interesting and thought-provoking.

The experiment follows the format of statistical experiments very well.

**Reasons To Reject:**

The scope of the claim implied by the paper's title is somewhat overstated. The study focuses on gender-based impression formation and its variation in LLMs. While I agree that the contribution is significant, I suggest that the claim in the title and framing of the paper be revised to match the actual scope of the work.

Although the paper discusses impression formation patterns in various social scenarios, the core of the analysis is limited to the interactions among Actor, Behavior, and Receptor (ABR). This focus is valid and appropriate, but I recommend framing the discussion more explicitly around ABR profiles, rather than suggesting a broader treatment of general "scenarios."

---

> ### Author Response · Authors · 2025-06-01
> **Response to Reviewer 7PD1's Comments**
>
> Thank you for your constructive feedback.
>
> **Regarding the scope and title:**
> While our framework is designed to be extensible to other types of social interactions (e.g., race, social status) and cross-cultural settings through alternative ACT dictionaries and impression formation equations, we agree that the original title and result statements overgeneralized the scope of this study. We will revise the title to:
> **“Impact of LLM Alignment on Impression Formation in Social Interactions: A Case Study of Gendered Impressions.”**
> For the camera-ready version, we will ensure that all framing and claims are revised to reflect our actual focus on gendered impressions within ABR (Actor–Behavior–Receptor) settings, with mentions of potential extensions to other domains.
>
> **Regarding the emphasis on ABR profiles:**
> We agree that ABR scenarios represent a limited class of social interactions. While the ACT framework does support richer modeling—including dimensions like emotions, traits, social status, situational context, and self-directed actions or object interactions—these are avenues for future expansion. We also acknowledge that many social situations involve more than two entities or more complex dynamics that go beyond the ABR structure.
>
> In response to your observation, we will revise the discussion section to explicitly clarify that our current study focuses on a specific subset of interactions and does not generalize to all types of social situations.
>
> **Regarding internal mechanisms:**
> In response to your excellent suggestion, we ran a preliminary analysis using Llama-3-Instruct. Just before generating the first output token (typically the predicted gender), the model attends more strongly to the actor than to the behavior (actor-to-behavior attention ratio = 1.12), suggesting the actor has greater influence on gender inference. Interestingly, this pattern shifts by valence. As shown below, attention to the behavior increases when the actor has low valence, indicating the model may rely more on behavior when the actor is perceived as “bad”:
>
> | Actor valence | Behavior valence | Attention weight ratio (Actor/Behavior) |
> |---------------|------------------|------------------------------------------|
> | High          | High             | 1.85                                     |
> | High          | Low              | 1.39                                     |
> | Low           | High             | 0.78                                     |
> | Low           | Low              | 0.65                                     |
>
> We will include this attention analysis in the final version to further support our discussion of internal representations and reasoning.
>
> We believe the revised framing, clarified scope, and new analysis of internal mechanisms—along with additional evaluations based on your and other reviewers' feedback (e.g., prompt variation and safety-trained model comparison)—substantially strengthen the paper. We welcome further thoughts during the discussion period and are happy to provide additional clarifications if needed.

---

> ### Comment · Reviewer_7PD1 · 2025-06-10
> **reply to author's response**
>
> Thank you for providing the preliminary analysis.
> I think this paper presents a valuable contribution.
> As I noted in my review, my main concerns were about the scope and framing of the paper.
> If these concerns are addressed as the authors have proposed, I would be in favor of acceptance.
> I would like to defer the final judgment to the meta-reviewer and will therefore keep my current score.

---

### Decision · Program_Chairs · 2025-07-08

**Decision:**

Accept

**Comment:**

This paper analyzes LLM dialogs as social interactions by comparison to the predictions of Affect Control Theory.

Pros:
- Results have broad interest outside the research community.
- Experiments are well-designed.
- ACT comparison is novel.

Cons:
- Needs more specific experimental support for claims about post-training failing at debiasing.
- ACT corpus sourced only from US